# Durability of Cellulosic-Fiber-Reinforced Geopolymers: A Review

**DOI:** 10.3390/molecules27030796

**Published:** 2022-01-25

**Authors:** Jie Liu, Chun Lv

**Affiliations:** 1College of Light-Industry and Textile Engineering, Qiqihar University, Qiqihar 161006, China; 01250@qqhru.edu.cn; 2College of Architecture and Civil Engineering, Qiqihar University, Qiqihar 161006, China

**Keywords:** cellulose fiber, plant fiber, geopolymer composites, durability, alkaline degradation, acid resistance

## Abstract

Geopolymers have high early strength, fast hardening speed and wide sources of raw materials, and have good durability properties such as high temperature resistance and corrosion resistance. On the other hand, there are abundant sources of plant or cellulose fibers, and it has the advantages of having a low cost, a light weight, strong adhesion and biodegradability. In this context, the geopolymer sector is considering cellulose fibers as a sustainable reinforcement for developing composites. Cellulosic-fiber-reinforced geopolymer composites have broad development prospects. This paper presents a review of the literature research on the durability of cellulosic-fiber-reinforced geopolymer composites in recent years. In this paper, the typical properties of cellulose fibers are summarized, and the polymerization mechanism of geopolymers is briefly discussed. The factors influencing the durability of cellulosic-fiber-reinforced geopolymer composites were summarized and analyzed, including the degradation of fibers in a geopolymer matrix, the toughness of fiber against matrix cracking, the acid resistance, and resistance to chloride ion penetration, high temperature resistance, etc. Finally, the influence of nanomaterials on the properties of geopolymer composites and the chemical modification of fibers are analyzed, and the research on cellulosic-fiber-reinforced geopolymer composites is summarized.

## 1. Introduction

Geopolymer is a kind of inorganic silico-aluminum cementitious material with a spatial structure prepared by the reaction of active low-calcium silico-alumina material with an alkaline activator, with a three-dimensional network composed of SiO_4_ and AlO_4_ tetrahedral unit structure. The synthesis of geopolymers requires active solid aluminosilicates and alkaline solutions containing alkali metals and silicates. Among them, the alkaline solution acts as a binder, alkali activator and dispersant [1]. Compared to cement-based composites, geopolymers have the advantages of a high early strength, fast hardening speed and wide range of raw materials [2]. Geopolymers have lower energy consumption and less pollutants in the production process, and they are considered to be the material with the highest potential to replace cement [3,4,5]. The concept of geopolymers was originally proposed to describe the inorganic aluminosilicate polymers synthesized with natural materials by French scientist Davidovits in 1978 [6]. His team used alkali metal silicate solutions to stimulate geological minerals to form polymeric aluminum silicate materials under strong alkaline conditions [7]. Subsequently, other solid silicate raw materials including fly ash [8,9], pozzolan [10], ground blast furnace slag [11] and other wastes [12,13,14] successfully prepared geopolymers.

Traditional cement-based composites have poor durability such as high temperature resistance and corrosion resistance. Geopolymer composites overcome this shortcoming well [15]. However, geopolymers are similar to ceramics, their flexural strength and tensile strength are poor, and they are very sensitive to microcracks. In order to solve the problem of the brittleness of geopolymers, the toughness of composites can be improved by incorporating fibers. Adding fibers to the geopolymer can limit the growth of cracks, and at the same time can enhance the ductility, toughness and tensile strength of the geopolymer [16,17]. In recent years, many scholars have conducted research on the durability of geopolymers. This type of research mostly focuses on the durability of geopolymers such as sulfate resistance, freeze-thaw, weathering, water absorption, abrasion resistance, alternating wet and dry effects, and chloride ion resistance [18,19]. By adjusting the ratio of silicon to aluminum, alkaline solution, curing conditions, and adding fibers, rice husk ash, etc., the mechanical properties and durability of the composites are improved. The presence of fibers improves the bending strength and fracture behavior of the material and promotes the toughening mechanism of the material. Fiber-reinforced geopolymers have better durability than cement-based materials of the same grade [20].

At present, the fibers used in composites mainly include natural fibers, metal fibers [21,22], inorganic fibers [23,24,25] and synthetic fibers [26,27,28]. Among them, there are many studies on synthetic-fiber-reinforced geopolymers, such as polyvinyl alcohol (PVA), polypropylene (PP), etc., but their production process pollutes the environment and faces difficulties in meeting the requirements of sustainable development [29,30]. Among natural fibers, plant or cellulose fibers (CFs) are most commonly used. Plant fiber is also called natural cellulosic fiber. It has the advantages of having a low cost, a light weight, strong adhesion, simple manufacturing process and biodegradability, which attract more and more scholars’ attentions [31,32,33].

In recent years, fiber-reinforced geopolymers have been studied from different perspectives. The effects of different types of fibers on the enhanced performance of geopolymer, and of cellulose fiber fabrics on the properties of cementitious composites and geopolymers were studied [34,35]. At present, there is no special review on the durability of plant- or cellulosic-fiber-reinforced geopolymer composites (CFGCs). The paper presents a review of the literature research on the durability of CFGCs for the past few years, and briefly summarizes the polymerization mechanism of geopolymers. Then, according to the performance characteristics of CFs, the factors that affect the durability of CFGCs are summarized and analyzed, including the degradation of CFs in the geopolymer matrix, the toughness of CF against matrix cracking, and the performance of acid resistance, anti-chloride ion penetration, high temperature resistance and so on. Finally, the addition of nanomaterials and the chemical modification of CFs affecting geopolymer composites are analyzed. However, there are few research cases about the freezing-thawing resistance and carbonization resistance of CFGCs, so this paper does not carry out in-depth discussion on relevant issues, and further relevant studies are needed in the future.

## 2. Typical Properties of CFs

CF is one of the most abundant natural resources in the world, and it is widely found in agricultural residues, such as rice straw, rice husk, maize straw, bagasse, wood shavings, wood chips, bamboo chips, etc. These agricultural residues are mainly composed of cellulose, hemicellulose, lignin, pectin, wax and some water-soluble materials. Cellulose is the most important component of CF, and its chemical formula is (C_6_H_10_O_5_)_n_. Cellulose is a macromolecular polysaccharide composed of glucose, which is a straight-chain polymer formed by linking countless D-glucopyranose anhydrides with β(1–4) glycosides, and its structure is regular and unbranched. Cellulose has a large number of hydroxyl groups on the molecular chain, which promote the formation of intramolecular and intermolecular hydrogen bonds [31]. CFs commonly used for geopolymer reinforcement include bast fibers, leaf fibers, stem fibers, etc. [35,36], as shown in Figure 1.

It can be seen from Table 1 that the density of CFs is roughly similar, with little difference, between 1.1 and 1.6 g·cm^−3^. The density of bast fiber is basically about 1.5 g·cm^−3^, its tensile strength is relatively large, and the tensile strength of fruit coconut husk fiber is relatively small.

## 3. Fiber-Reinforced Geopolymer Composites

Geopolymers can be prepared in two ways: alkali excitation and acid excitation. According to the different active raw materials, alkali excitation methods are mainly divided into alkali-silicate glass body cementing materials and alkali-silicate mineral cementing materials. Alkali-silicate glass body cementing materials mainly use amorphous silicate glass bodies as raw materials, such as slag, fly ash, various metallurgical slags, coal gangue, etc., and the main raw alkali-silicate mineral cementing material is a crystalline mineral, such as clay, feldspar and other tailings.

### 3.1. The Polymerization Mechanism of Geopolymer

Under the condition of strong alkali, the silicon-oxygen bonds and aluminum-oxygen bonds of active materials such as kaolin are broken to form oligomers of polymer monomers, namely oligomeric silicon-oxygen tetrahedra and aluminum-oxygen tetrahedra. Under the same conditions, oligomeric silicon-oxygen tetrahedrons and aluminum-oxygen tetrahedrons are dehydrated and polymerized to form geopolymers with a three-dimensional network structure in space [44]. It is generally believed that the reaction of geopolymers can be divided into four processes: dissolution, diffusion, polymerization and solidification. Using metakaolin as the active material and (NaOH) or (KOH) as the alkali activator, the reaction mechanism of the resulting geopolymer is shown in Figure 2 [45,46].

It can be seen from Figure 2 that the aluminosilicate raw materials (precursors) gradually dissolve in (NaOH) or (KOH) alkali activator, producing a large amount of silicon and aluminum monomers. These monomers gradually diffuse in the solution from the surface to the inside, and quickly undergo a polycondensation reaction to form silico−alumina oligomers. The oligomer gel phase solidifies and hardens to form geopolymers.

### 3.2. Fiber Matrix Interface Bonding Mechanisms

A geopolymer composite is composed of fiber and a matrix with different properties, and the interface between the fiber and matrix is formed. The interface of the composite includes the geometric surface of the matrix and the fiber in contact with each other and the transition area, which is an extremely complex microstructure. Adjusting the bonding state of the fiber and the matrix interface, and optimizing the characteristics of the interface layer between the fiber and the matrix can make the geopolymer composites achieve the best performance. Improving the interfacial adhesion between the fiber reinforcement and the matrix is the most critical factor in the interface control technology of composites. The bonding forms of the fiber and matrix interface generally include interdiffusion, electrostatic adhesion, chemical bonding and mechanical interlocking [46,47]. According to the microscopic morphology of the bonding of fibers and geopolymers, the interface bonding is usually mainly in the form of mechanical interlocking.

## 4. Research Status of the Durability of CFGCs

Durability refers to the ability of a material to resist the long-term destructive effects of both itself and the natural environment. Generally, the better the durability of a material, the longer its service life will be [48,49]. At present, scholars have conducted a lot of research on the properties of CFGCs, such as crack resistance, acid corrosion resistance, chloride ion penetration resistance, dry and wet cycle, and high temperature resistance. Microscopic analysis shows [50] that geopolymers can form an impermeable layer under the action of fibers, making the geopolymer matrix structure more compact. The good adhesion between the fiber and the matrix enables the geopolymer composite to prevent crack propagation, resist freeze-thaw and penetration erosion, and enhance its durability.

### 4.1. The Alkaline Degradation Mechanism of CFs

The durability of CFGCs involves the durability of the matrix and the durability of the CFs in the matrix. As we all know, the amorphous components of CFs in cement concrete will be degraded to varying degrees in an alkaline environment [36]. Similarly, on the one hand, the fiber has good compatibility with the polymer matrix, on the other hand, the fiber in the matrix also degrades [51,52]. First, lignin and part of the hemicellulose were degraded, and then the hemicellulose was completely degraded, destroying the integrity and stability of the CF cell wall. This results in the peeling of the cellulosic fiber from the CF cell wall and the failure of the cellulose fiber, which leads to the complete degradation of the CF. Due to the alkaline hydrolysis of cellulose fiber chains, hemicellulose and lignin, the integrity of the fiber and geopolymer matrix interface area is lost, thereby damaging the mechanical properties and durability of CFGCs.

Geopolymer-based materials composed of cellulose hemicellulose and lignin have unique microstructures. A total of 5% mass content of cellulose hemicellulose and lignin could effectively improve the flexural and compressive strength of geopolymer [53]. In fact, the increase in the content of lignin and hemicellulose leads to the porosity, low density and brittleness of the composite, which reduces the flexural and compressive strength of the composite; the alkaline degradation of hemicellulose reduces the degree of polymerization of the composite. However, as the cellulose content increases, the matrix structure becomes denser, with fewer pores, and the toughness of the composite increases. The geopolymer matrix and the cellulose fibers also showed good bonding without significant degradation.

Different types of CFs and different external environmental conditions make the degree of degradation of cellulosic fibers different, and also have great influence on the durability of composite materials. The flax-fabric-reinforced composite was immersed in water, seawater and 5% sodium hydroxide alkaline solution. After aging for one year, the tensile and bending properties of the composite were tested. The results showed that the degradation of the composites was most serious in 5% sodium hydroxide alkaline solution [54]. After impregnating bamboo pulp and nanocellulose fibers with cement concrete and geopolymer, lignin was removed from the fiber surface, and hemicellulose and cellulose were degraded to a certain extent. The tensile strength of pulp sheet decreased by 70% and 34%, respectively [55]. Due to the inherent properties of CFs, although geopolymers did not contain calcium hydroxide, the high alkalinity of the slurry also sped up the degradation process. The mineralization and partial degradation of hemicellulose of black locust and longleaf acacia grains and bagasse were found in the geopolymers, indicating that the durability of black locust and longleaf acacia grains in alkaline substrates deteriorated [56].

Although the degradation degree of CF in the geopolymer matrix is relatively weaker than that in cement, the degradation of the alkaline matrix of the geopolymer also affects the durability of CFGC.

### 4.2. Crack Resistance and Toughness of CFGCs

In recent years, there have been many reports on the physical, thermal and mechanical properties of CFGCs, but few studies on their durability. The current research on the durability of geopolymers mainly focuses on the toughening and cracking resistance of materials, the resistance to sulfate erosion, resistant to high temperatures, chloride ion corrosion resistance, frost resistance, etc. CFs can inhibit and stabilize the development of micro-cracks in the geopolymer, which is an effective way to alleviate the performance degradation of geopolymer composites. As we all know, the durability of geopolymer composites is closely related to its compactness and crack resistance, and good toughness helps to improve the durability of geopolymers.

#### 4.2.1. The Effect of Bast Fiber on the Toughness of CFGCs

Most bast fibers have good strength and are widely used in the manufacture of ropes, twine, packaging materials and industrial thick cloth. Bast fibers mainly include hemp, flax, jute, ramie and kenaf. Tests show that most bast fibers have a good strengthening and toughening effect on CFGCs.

Hemp fiber has a positive effect on the microstructure and fracture structure of the geopolymers and can effectively improve the mechanical properties of the composites. The matrix with relatively brittle initial strength shows a higher increase in relative toughness. Eyerusalem et al. [57] used randomly oriented hemp as a reinforcement fiber and observed the fiber pulling out after loading, which showed the toughening ability of fiber. The tensile strength of the composite reinforced by 9% hemp fiber by volume was especially improved, and its tensile strength was about 5.5 MPa. At the same time, the addition of hemp fiber also slightly affected the density water absorption, compressive strength and flexural strength of the composite, but significantly improved the energy absorption capacity of the composite [58]. With the increase of fiber content, the compressive strength of the composite decreases continuously, indicating that the fiber distribution in the matrix becomes more and more uneven.

Bast fibers can change the initial brittle behavior of the geopolymer matrix to make it a ductile material, and their arrangement direction and treatment mode in the matrix also affect the toughening effect of composites. Trindade et al. [59] adopted a one-way and bidirectional arrangement of sisal and jute as reinforcement materials and showed that the composite material exhibits strain and flexural hardening behavior under tensile and bending action and produces multiple cracks. Sáez-Pérez et al. [60] found that under the two experimental conditions of fresh and wet storage for 6 months, similar to other pretreatment methods to improve the properties of hemp fiber, wet storage caused an increase in cellulose content and improved the mechanical properties of the geopolymer. Na et al. [61] soaked alkali-treated Kenaf fiber in 1 mol/L CaCl_2_ solution, which improved the compatibility between fiber and matrix, increased the flexural strength by 69.1%, and increased the toughness by 473%. After loading, the fiber had a typical failure mode of toughness.

The effects of different content of bast fiber on the properties of the geopolymer composites are different. Assaedi et al. [62] used flax fiber as reinforcement material to significantly improve the flexural strength, compressive strength hardness and fracture toughness of geopolymers.

Figure 3 shows typical stress–strain curves of pure geopolymer and composites with different fiber content. It can be seen that the composite with a fiber content of 4.1 wt% has the highest flexural strength among all composites. The flexural strength of the composite has been increased from 4.5 MPa for pure geopolymer to 23 MPa. It shows that increasing the content of flax fiber can significantly improve the flexural strength of the composite.

However, the fiber content should not be too high. Korniejenko et al. [63] also showed that the addition of flax fiber led to the decline of mechanical properties of composite materials. When flax fiber was added at 8%, the compressive strength of the material decreased by about 50%. Similarly, the bending strength of the pure geopolymer matrix was 3.45 MPa, and that of 8% flax-fiber-reinforced composite was 2.13 MPa. The results show that the fracture toughness of the composite increased the most when the flax fiber content was 4.1 wt%.

Generally speaking, fiber debonding and bridging slow down the crack propagation of composites and increase the fracture energy. The fracture toughness of the composite containing flax fiber is significantly higher than that of pure polymer, and the higher the fiber content, the higher the fracture toughness. This enhancement is due to the unique anti-breaking ability of flax fiber, which leads to increased energy dissipation of fiber matrix interface crack deflection, fiber debonding, fiber bridging, fiber pull-out and fracture, as shown in Figure 4a–d. It can be seen that the geopolymer adheres to the surface of the fiber, showing good adhesion between the fiber and the matrix. SEM images show various toughening mechanisms including crack bridging fiber pulling out and fiber fracture. Because of the degradation of flax fiber, flax-fiber-reinforced geopolymer exhibits higher net weight loss than pure geopolymer.

It can be seen that bast fibers have a good toughening effect on CFGCs, which further reduces the possibility of CFGCs cracking, thus effectively improving their durability.

#### 4.2.2. The Effect of Leaf Fiber on the Toughness of CFGCs

Leaf fibers are vascular bundle fibers obtained from the leaves of monocotyledonous plants. There are many varieties of leaf fiber, including raffia fiber, pineapple leaf fiber, sisal fiber, abaca fiber, agave fiber and so on. Similar to bast fibers, leaf fibers are often used for CFGCs, among which sisal-fiber-reinforced geopolymers are more studied.

The properties of the composites can be affected by the different content and length of leaf fiber. Ampol et al. [64] blended sisal fiber and coconut fiber into geopolymers with volume fractions of 0%, 0.5%, 0.75% and 1.0%, tested the mechanical properties of the geopolymers, and combined them with glass fiber. The results showed that, compared with glass fiber, adding sisal fiber and coconut fiber as reinforcement materials significantly improves the tensile and flexural strength properties. At the same time, the processing performance, dry density, ultrasonic pulse speed and compressive strength values all have a tendency to decrease. Zulfiat et al. [65] conducted compressive strength tests on pineapple fibers with lengths of 10, 20 and 30 mm, and geopolymers with fiber weight percentages of 0, 0.25 and 0.50%, respectively. The composite with a fiber content of 0.50% and a fiber length of 30 mm had a compressive strength of 41.468 MPa and a maximum bending strength of 9.209 MPa. Studies have shown that the compressive strength and flexural strength of the geopolymer mortar reinforced by 0.5% by mass pineapple fiber is higher than that of the geopolymer reinforced by 0.25% by mass pineapple fiber.

However, whether it is leaf fiber or bast fiber, the excessive fiber content leads to the disharmony between the fiber volume and the matrix volume, which reduces the mechanical properties of the geopolymer, indicating that the appropriate amount of fiber can improve the mechanical properties of the geopolymer.

#### 4.2.3. The Effect of Seed Fiber on the Toughness of CFGCs

Seed fibers are single-cell fibers grown from epidermal cells of plant seeds. Mainly include cotton fiber, kapok fiber and so on. Cotton fiber is amongst the most well-known of CFs. Cotton fiber has unique properties such as high cellulose content, good moisture absorption, excellent heat resistance, light resistance, alkali resistance and higher breaking strength, so it can play an important role in geopolymers. Compared to the pure geopolymer, the addition of cotton fiber gradually improves the fracture toughness of the fiber-reinforced geopolymer composite. Cotton fiber has the characteristics of energy absorption through fiber fracture, fiber matrix interface debonding, fiber pull-out and fiber bridging, etc., which slow down the propagation of cracks and increase the fracture energy, thus playing an important role in enhancing the toughness of the matrix [66]. Tests have shown that the addition of cotton, sisal or coir fiber composites can improve its bending properties [67].

However, it should be noted that the increase in the volume of the hydrophilic natural fibers in the geopolymer matrix will adversely affect the strength of the composite. The fly-ash-based geopolymer composite reinforced with cotton fabric can prevent the cotton fabric from degrading at high temperatures. When the fabric is arranged in a horizontal direction with respect to the applied load, it achieves a higher load and greater deformation resistance than a vertically arranged fabric [68,69]. The results show that cotton fabric orientation affects the bending strength, compressive strength, hardness and fracture toughness of geopolymer composites.

#### 4.2.4. The Effect of Fruit Fiber on the Toughness of CFGCs

Fruit fiber refers to fiber obtained from the fruit of a plant. It is mainly composed of cellulose and associated biomass and intercellular substance, such as coconut fiber. Kroehong et al. [70] found that the addition of oil palm fiber had a significant impact on the physical and mechanical properties and microstructure of geopolymer with high calcium fly ash. The increase of oil palm fiber content reduced the compressive strength of geopolymer but improved the bending strength and toughness of the material and changed the failure behavior of the composite. In addition, with the increase of fiber content, the pore size and total porosity of the material increased, while the thermal conductivity decreased.

In fact, the strength and deformation curve of the fruit fiber is similar to that of the bast fiber and generally the strength value of the fruit fiber is lower than that of the bast fiber. Mazen [71] used loofah fiber as a geopolymer reinforcement material. Compared with pure geopolymer, the compressive strength of the composite was increased from 13 to 31 MPa, and the bending strength was increased from 3.4 to 14.2 MPa. After 20 months of aging time, the flexural yield strength of the composite with 10% loofah fiber content increased from 8.6 to 9.8 MPa, as shown in Figure 5. The increase in yield strength is due to the continuous polymerization of geopolymers as the aging progresses. During the 20-month aging period, the ultimate flexural strength, strain hardening and flexural modulus all changed slightly. The aging study showed that the mechanical properties of the composite material did not decrease significantly within 20 months.

In another study, Gabriel et al. [72] synthesized wood fiber-reinforced geopolymer composites with fly ash, sand and wood fibers, and added 5, 10, 15, 20, 25, 30 and 35 wt% variable wood fibers. As the amount of wood fiber added increased, the mechanical properties decreased [73]. Su [74] used fly ash and slag as raw materials to prepare geopolymers and improved its crack resistance and strength by adding lignin fibers, polypropylene fibers and alkali-resistant glass fibers. When the fiber addition amount was 0.75%, the strength of fiber-reinforced geopolymer was the best. The coherence between natural fiber and geopolymer matrix is lower than that of artificial fiber. The improvement effects of fiber reinforcement and shrinkage resistance are, in order, PP fiber, alkali-resistant glass fiber and lignin fiber. The fiber not only prevents the separation from the geopolymer matrix, but also inhibits the generation and expansion of cracks, and ultimately improves the strength of the fiber.

#### 4.2.5. The Effect of Stem Fiber on the Toughness of CFGCs

Agricultural waste straws are mostly stem fibers, such as rice, wheat, sorghum, bagasse, and so on. Chen et al. [75] reported that when the fiber content was less than 2.0%, the increase in the content of sweet sorghum fiber in the geopolymer caused the density and unconfined compressive strength of the composite to continue to decrease, while the bending strength, tensile strength and peak toughness had been significantly improved. It shows that the main function of fiber is not to improve the compressive strength of composites, but to improve its flexural performance and control the further development of matrix concrete cracks.

#### 4.2.6. The Effect of Grass/Reeds Fiber on the Toughness of CFGCs

Grass/reeds fiber includes reed, bamboo fiber, corn fiber, and so on. Kaushik et al. [76] used 5 wt% of untreated bamboo fiber to reinforce potassium-based metakaolin geopolymer to obtain a four-point flexural strength of 7.5 MPa.

As mentioned above, cellulosic fiber is the main source of toughness of CF reinforced geopolymers, whether it is bast fiber, fruit fiber, leaf fiber, seed fiber, wood fiber or grass fiber. The strength and fracture resilience of CFGCs with different fiber contents are quite different. Relevant experiments show that, compared with pure geopolymer, the fracture toughness of geopolymer containing 2% cellulosic fiber can be increased 4-fold [77,78].

The microstructure analysis shows that there may be chemical interaction between organic fiber and inorganic polymer chain, and the failure dynamics of geopolymer matrix composites include crack bridging, fiber pulling out and fiber tearing mechanism [62]. Compared with other fibers, CFs have higher specific modulus and elongation at break, and are more evenly distributed in the geopolymer matrix. The presence of CFs usually increases the tensile strength, flexural strength and toughness of the geopolymer composite, thereby improving the durability of the geopolymer composite. On the other hand, the use rate of CF should not be too high because too much CF will increase the porosity of the geopolymer composite, make the fiber and the matrix poorly bonded, and fail to achieve the desired strengthening and toughening effect.

### 4.3. Resistance to Sulfate Attack of CFGCs

Sulfate resistance is one of the important indexes of cement-based material durability. As a new cementing material which can replace the traditional Portland cement, geopolymer has better sulfate resistance than cement. After sulfate solution erosion, the mechanical strength, microstructure and surface morphology of CFGCs will change to some extent.

Fan et al. [79] used sisal fiber and polyvinyl alcohol fiber (PVA) to strengthen metakaolin-based geopolymer, indicating that fiber incorporation can greatly improve the physical properties of the geopolymer, while fiber doping can improve the sulfate erosion resistance of the geopolymer. Compared with the low concentration of sulfate, the high concentration of sulfate erodes the geopolymer with more cracks and pores, and the compressive strength decreases more obviously. The results showed that the fiber-reinforced geopolymer prepared by 0.5 wt% PVA and 0.75 wt% sisal fiber had the highest compressive and flexural strength and was the most stable.

The microstructure of the fiber-reinforced metakaolin-based polymer samples after curing for 28 d and after being eroded by sulfate at a concentration of 5 wt% and 15 wt% for 28 d is shown in Figure 6 [79]. With the increase of sulfate concentration, it can be seen that there are certain cracks and holes. This is because the sulfate gradually infiltrates into the geopolymer sample during the erosion process, occupying some voids in the geopolymer. The accumulation continues, causing the development of cracks.

Similar to the above, cellulosic geopolymer composites can be stabilized by acid rain leaching over long periods of time. Jin et al. [80] mixed straw with a mass fraction of 4% into the geopolymer, and the compressive strength of the composite was greater than 30 MPa. Under acid rain leaching with a pH value above 3, the compressive strength was maintained at about 36 MPa, and the acid rain leaching resistance was good. When the composite was immersed in thiobacillus thioxide, the compressive strength of the composite was 26.3 MPa for 21 days and 18.4 MPa for 28 days. The results showed that although thiobacillus thioxide had a certain effect on the compressive strength of the straw geopolymer, its amorphous three-dimensional network silicoaluminate structure still existed.

However, in a strong acid solution, the properties of CFGCs also change greatly. Jin [81] synthesized a metakaolin-based geopolymer from rice straw fiber. The straw geopolymer was soaked in sulfuric acid solution at pH 1.0, 3.0 and 5.0 and sodium hydroxide solution at pH 9.0, 11.0 and 13.0. After being immersed in an acid solution with a pH of 1.0 and an alkali solution with a pH of 13.0 for 28 days, the compressive strength of the samples decreased from 50.28 to 28.90 MPa and 40.00 MPa, respectively. Soaking the sample in a sodium hydroxide solution with a pH of 9.0 had the best effect.

The durability of CFGCs can be judged by soaking in sulfuric acid solution to test its cross-section reduction, weight loss, compressive strength loss and other changes. Maan et al. [82] prepared several concrete mixtures by using fly ash and crushed palm oil clinker (POC) as lightweight aggregate and oil palm trunk fiber (OPTF) as natural fiber reinforcement. In the concrete mixture, POC was added to the mixture in proportions of 25%, 50%, 75% and 100% by mass, and OPTF was added to the mixture in proportions of 1%, 2% and 3% by volume, respectively. The results showed that when the compressive strength was above 30 MPa, the optimal replacement rate of POC content was 75%, and OPTF content was 1%. The addition of POC and OPTF reduced the acid resistance of concrete.

Table 2 shows the strength and water absorption test results of the composites in 28 days. The addition of POC significantly improved the water absorption of the mixture. Due to the porosity of POC, the water absorption value increased by one-third when POC content was 25%, and 4.3-fold when it was completely replaced, as shown in Table 2. The water absorption value of geopolymer increased exponentially when added to OPTF. When the OPTF content was 1%, the increase was about two-thirds. When the content of OPTF was 3%, the increase was 3.1-fold. Through the durability test of immersion in sulfuric acid solution, the section reduction, weight loss and compressive strength loss have no obvious change, indicating that the corrosion resistance of sulfuric acid is good.

### 4.4. Resistance to Chloride Ion Penetration of CFGCs

The durability of CFGCs is an important property in engineering application. In fact, through the study of the long-term durability of fly-ash-based geopolymer exposed for 10 years under deep burial and complete saturation conditions in a severe salt lake environment, it was found that [83], compared with cement concrete, geopolymer had an adverse effect on chloride ion transport, with higher chloride ion diffusion coefficient and lower bond ability.

Zhou et al. [84] conducted a hydrochloric acid erosion test on biogeopolymer of cotton stalk powder and found that the addition of untreated cotton stalk fiber reduced the density and compressive strength of geopolymer, while the flexural strength slightly increased. The compressive strength and flexural strength of cotton stalk fiber after alkali treatment were 4.8% and 11.5% higher than those without alkali treatment, respectively. The treated cotton stalk powder could effectively improve the compressive strength of geopolymer but reduce the acid corrosion resistance of geopolymer. Through analysis, the effect of cotton stalk powder on geopolymer is mainly filling and cementation. The sugar precipitated from cotton stems in an alkaline environment reduces the compactness of the geopolymer gel.

The effects of different concentrations of hydrochloric acid on the resistance of CFGCs are very different. Ribeiro et al. [85] immersed bamboo fiber geopolymers in sulfuric acid and hydrochloric acid of 0, 5 and 15 wt% for 7, 28 and 112 days to study their appearance, quality changes and compressive strength behavior. No mass loss was observed in 0% acid (100% water), indicating durability in water. The mass loss of geopolymer increased from 5, 10 and 15 wt% to 2.7%, 3.5% and 4.4%, respectively, with the increase of acid concentration.

Figure 7 shows the microscopic morphology of bamboo-fiber-reinforced geopolymer before and after chloride ion erosion. (a) shows SEM images of 4.1 wt% bamboo-fiber-reinforced geopolymer soaked in 15 wt% hydrochloric acid for 28 days. The phenomenon of micropores formed by acid leaching is obvious. (b) shows the parallel periodic micro-cracking in the geopolymer matrix. (c) shows the undissolved chopped bamboo fiber in 15 wt% sulfuric acid treatment for 28 days. (d) shows the formation of periodic parallel microcrack complexes in bamboo-fiber-reinforced geopolymer.

### 4.5. Performance of CFGCs against Wetting/Drying Cycles

The performance of composites against wetting/drying cycles is an important part of durability. Trindade et al. [86] formulated two geopolymers of 100% metakaolin (MK) and 60% (MK) + 40% blast furnace slag (BFS). The mechanical behavior of the two matrices is changed by the jute reinforcing fiber to make it ductile and change its crack mode. Jute fiber promotes the formation of (C-A-S-H) gel and significantly improves the compressive strength of the material. The five-layer jute-fabric-reinforced composite exhibits strain and flexural hardening behavior under tension and bending, and produces multiple cracks. It can be seen in Figure 8 that after 15 wetting/drying cycles, the first crack strength values of the two composites decreased significantly, but the ultimate strength did not change significantly. This behavior shows that after 15 wetting/drying cycles, the ultimate mechanical capacity of the composite does not change significantly. The composites all exhibit deflection hardening behavior, which leads to smaller crack openings, and various cracks are formed after accelerated aging. The fibers did not degrade significantly after 15 wetting/drying cycles, indicating that the jute fabric-reinforced geopolymer has superior durability compared with Portland cement matrix composites.

A similar report was also found in ref. [87], Nkwaju et al. used iron-rich red clay and bagasse fibers as raw materials to prepare geopolymer composites, and found that the addition of fibers facilitated the transition of the fracture behavior of geopolymers from brittleness to toughness. With the increase of fiber content, when the fiber mass fraction was 3%, the elastic modulus increased by 50%. After 20 wetting/drying cycles, the performance of the geopolymer composite material had been improved, and the ductility had been improved. It shows that the wetting/drying cycles improves the fiber matrix bond, thereby increasing the ductility of the composite. Santos et al. [88] evaluated durability by accelerating aging through 10 wetting and drying cycles. The composite (0 cycles) was about 15 MPa in the bending test, and the aged composite reached 11 MPa, indicating that the wetting and drying cycles had good durability. The fiber of the composite that had been naturally aged for 3 years has almost no degradation, and the composite had good durability.

The mechanical behavior of the composites at the inelastic stage, such as cracking mechanism, strength and ductility, was tested by a bending test. Canpolat et al. [89] studied the influence of wetting-drying curing system on the performance of fiber reinforced metakaolin-based geopolymer composites. Similarly, Asante et al. [90] also found that the specific strength of the pine and eucalyptus particle geopolymer composite material was reduced by 15.32% after multiple soaking and drying.

### 4.6. High Temperature Tolerance of CFGCs

Compared with cement-based materials, geopolymer materials have better durability than cement-based materials. Alomayri et al. [91] tested a geopolymer composite material containing 0.83% wt% cotton fabric by exposing it to high temperatures of 200 °C, 400 °C, 600 °C, 800 °C and 1000 °C. As the temperature increased, the compressive strength, flexural strength and fracture toughness of geopolymers all decreased. The high temperature severely degrades the cotton fiber, resulting in holes and small channels in the composite material, which makes the composite material exhibit brittle behavior. When the temperature reaches above 600 °C, the mechanical properties of the composite material are significantly reduced. Alomayri et al. [92] used cotton-fabric-reinforced polymer composites to characterize their thermal properties through thermogravimetric analysis, and evaluated their mechanical properties, such as flexural strength, fracture toughness, flexural modulus and impact strength. When the fiber content was 2.1 wt%, the mechanical properties of the fiber were improved. The thermal analysis results showed that the fly-ash-based polymer can prevent the degradation of cotton fabrics at high temperatures. Amalia et al. [93] studied the high temperature resistance of fly-ash-based hybrid geopolymers with pineapple leaf fibers as aggregates, and the results showed that pineapple leaf fibers had great potential in geopolymer reinforcement or lightweight aggregates.

Another experiment [86] found that there were fine crack networks on pure geopolymer heated at 400 and 600 °C. After over 600 °C, severe cracks appear on the surface. When cotton fiber was added, no cracks were found on the surface of geopolymer at the same temperature. It shows that cotton fiber is very effective in preventing the matrix from forming cracks at high temperature. The structure of the composite becomes more porous, and the expanded water vapor escapes without causing major damage to the microstructure. This degradation of cotton fibers will facilitate the behavior of geopolymers under heat exposure. The porosity and small channels produced by the degradation of cotton fibers can reduce the internal vapor pressure, thereby reducing the possibility of cracking.

Compared to ordinary Portland cement, geopolymer has better acid resistance and sulfate resistance, but its resistance to carbonization is slightly worse. Frost resistance is also an important performance for the durability of plant-fiber-reinforced geopolymers, but there are few studies in this area. Xuan et al. [94] prepared geopolymer-based CF composites using industrial waste slag and agricultural and forestry residue bagasse as raw materials. The bending strength was analyzed, and its appearance and microstructure were analyzed. It shows that CFGCs have better frost resistance.

## 5. Other Factors Affecting the Durability of CFGCs

In general, there are basically two ways for improving the durability of CFGCs. One way is to add nanomaterials into the matrix; the other way is to modify CFs.

### 5.1. The Effect of Nanomaterial Addition on the Durability of CFGCs

Pore structure is the basis of the theory of geopolymer mix ratio design and its relationship with mechanical properties and is closely related to the macro-mechanical properties and durability of geopolymers. Nanomaterials have unique nano-effects such as volume effect, surface effect, quantum size, quantum tunnel, etc., which lead to unique physical and chemical properties of nanomaterials and nanostructures. Incorporating nano-SiO_2_ can not only speed up the polymer polymerization reaction process, in which the unreacted nano-SiO_2_ particles are wrapped by the geopolymer, but also play the role of particle filling, so that the overall structure of the composite material is more compact, and the polymerization is improved, as is the durability of objects [95].

Generally, CFs will deteriorate to varying degrees after being exposed to an alkaline environment. The attack of alkali ions leads to the weakening of cellulose and hemicellulose, and the mineralization of fiber cell walls in the geopolymer pulp leads to fiber brittleness. Assaedi [96] evaluated that after 32 weeks of aging, the flexural strength of geopolymer composites decreased. Figure 9a,b show the load deflection behavior of fiber-reinforced geopolymer (GP/FF) composites and nano-SiO_2_ fiber-reinforced geopolymer (GPNS-1/FF) composites at 4 and 32 weeks, respectively. Among the two composites, the composite containing nano-SiO_2_ has a higher load capacity.

The flexural strength of the flax-fiber-reinforced geopolymer composite material was reduced by 22.4%, while the flexural strength of GPNS-1/FF was reduced by approximately 10.3%. It shows that after 32 weeks of aging, the flexural strength of the nanocomposite decreases less. The analysis shows that nano-silica consumes the alkaline solution and reduces the alkalinity of the system, thereby reducing the degradation of flax fibers. In addition, nano-silica accelerates the geopolymer reaction and increases the geopolymer gel in the matrix. The amount of it improves the density of the matrix and actually improves the adhesion between the fiber and the geopolymer matrix, thereby enhancing the durability of the geopolymer.

In a similar way, Saulo et al. [97] improved the compressive strength and stiffness of geopolymers for 7 days by adding microcrystalline cellulose fibers. After 28 days, due to the degradation of microcrystalline cellulose, the geopolymer reduced its mechanical properties. The group of Cut [98] studied the effect of different cellulose nanocrystal concentrations on the mechanical properties of geopolymers. Lower concentrations of cellulose nanocrystals (<0.5%) could produce higher strength geopolymers. The higher concentration of cellulose nanocrystals prevents the pyrolysis of the geopolymer in an unstable solidification environment and enhances the corrosion resistance of the composite. Rahman et al. [99] studied the synergistic effect of silicon dioxide and silicon carbide whiskers derived from rice husk ash. It showed that the spherical silica nanoparticles prepared from rice husk ash reduced the nanoporosity of the geopolymer by 20% and doubled the compressive strength. When rice husk ash and silicon carbide whiskers were added at the same time, the flexural strength increased by 27% and 97%, respectively. The increased compressive strength of silica nanoparticles is related to the decrease of porosity, and silicon carbide whiskers can effectively improve the bridge network and crack resistance.

Assaedi et al. [100] synthesized a geopolymer composite material reinforced by flax fabric and nano-clay flakes, and added nano-clay flakes to strengthen the geopolymer matrix at 1.0, 2.0 and 3.0% by mass. The 2.0 wt% nanoclay had the best effect, increasing the density and reducing the porosity, thereby increasing the flexural strength and toughness. Rahmawati et al. [101] used Typha as a new raw material for separating cellulose nanocrystals and extracted cellulose from stem fibers by alkaline and bleaching methods. Then, cellulose nanocrystals were separated from the extracted cellulose by acid hydrolysis. The acid hydrolyzed cellulose nanocrystal had good thermal stability at 240 °C, which was higher than that of raw meal. Cellulose nanocrystals have the potential as a geopolymer cement reinforcement agent.

Based on the above studies, it can be concluded that nanomaterials can be used not only as a filler to improve the microstructure of the binder, but also as an activator to support the geopolymer reaction and produce a higher content of geopolymer gel. This enhances the adhesion between the geopolymer matrix and CF, thereby improving the properties of CFGCs.

### 5.2. The Effect of Fiber Modification on the Durability of CFGCs

Compared with other types of fibers, some limitations of CFs, such as biodegradation, UV degradation or weak bonding, affect their mechanical properties and durability. In order to improve the performance of CFs in the geopolymer matrix, a modification treatment method is usually used. Alkalization is one of the chemical modification techniques of bio-based materials. The purpose of the treatment is to have less impurities in the fiber and increase the adhesion of its contact surface.

Scholars have many research cases on the modification of bast fibers. Kumar et al. [41] used 10% (NaOH) solution for alkali treatment of ramie with a maximum treatment temperature of 160 °C and obtained strong fibers with low lignin content and good fiber separation. Lazorenko [102] et al. treated alkaline media with 5% (NaOH) solution mercerizing and ultrasonic (22 kHz, 500 W). The combined treatment of alkali and high-intensity ultrasound is an effective way to treat and modify fiber-reinforced polymer composites, which has the best technical and economic effects. Maichin et al. [103] studied the influence of sodium hydroxide concentration on the pretreatment performance of hemp fiber and the self-treatment behavior of hemp fiber in geopolymer composites. The self-treatment behavior of hemp fiber in the geopolymer can improve the final performance of the hardened product. The self-treatment process is to add fibers to the geopolymer mixture without any pre-alkaline treatment. Similarly, Maichin et al. [104] also discussed the influence of geopolymer alkalinity on fiber self-treatment. The results showed that the self-treatment process is controlled by the alkaline environment in the geopolymer system. After the fiber is alkalized, the surface of the fiber is modified, so that the fiber has stronger cohesiveness and better compatibility with cement paste. Pickering et al. [38] discussed different methods of chemical modification of hemp fiber to make it have good water repellency, chemical resistance and good mechanical properties. Georg et al. [105] used pretreatment and surface modification to remove short hemicellulose, lignin, pectin and wax, increase the interface adhesion between the matrix and flax fibers, and optimize the rheology of the geopolymer slurry performance. Roy et al. [106] applied different chemical treatments to the abaca fiber to change its surface characteristics and improve the adhesion to the fly-ash-based polymer matrix. It showed that the tensile strength of abaca fiber without alkali pretreatment and soaked in (Al_2_(SO_4_)_3_) solution with pH6 for 12 h had the highest tensile strength. Chemical treatment and deposition of aluminum compounds make the surface rougher. This improves the interfacial bonding between the geopolymer matrix and the fibers, while the geopolymer protects the treated fibers from thermal degradation.

Sisal is typical of leaf fiber. Figure 10 shows the microscopic morphology of sisal fiber-reinforced metakaolin-based polymer. It can be observed that the fiber is broken or torn at the section, as shown in Figure 10a,b. The phenomenon of the fibrillation of sisal fiber indicates that the sisal fiber plays a role in bearing force during the crack propagation and fracture process of the sample, which causes the damage and tear of the fiber. As shown in Figure 10c,d, the interfacial bonding ability between the alkali-treated sisal fiber and the metakaolin-based polymer material has been significantly improved. It can be seen that the sisal fiber breaks under external force. The fiber is not drawn out due to external force, but breaks during the stress process, indicating that the interface bonding force between the sisal fiber and the geopolymer is greater than the maximum stress that the fiber can withstand, which shows that the alkali treatment can greatly improve the interface bonding between sisal fiber and metakaolin-based polymer material.

The modification treatment of stem fiber is mainly alkali treatment. Huang et al. [107] found that both untreated and alkali-treated rice straw can significantly increase the flexural strength of geopolymers. The bonding effect of the geopolymer matrix and straw after alkali treatment was better than that of untreated rice straw. During the curing time of 28 days, the flexural strength of the alkali-treated straw-reinforced geopolymer composite with a fiber content of 10% reached 13.6 MPa. Workiye et al. [108] chemically treated corn stalks with 98% pure sodium hydroxide for 30 min, and prepared 0, 0.1, 0.2, 0.6 and 1% by mass corn stalk monocellulose-reinforced geopolymer composites. The material indicates that the proper addition of single fiber of corn stover can improve the compressive strength of the calcined kaolinite base polymer.

Ribeiro et al. [109] found that bamboo fibers and strips enhance the compressive strength of geopolymers. The alkali-treated micro-bamboo fiber had a compressive strength of 23–38 MPa, which was lower than the 56 MPa of pure geopolymers, but still had good structural applications. Alkali treatment and water treatment have no significant difference in the bending strength of bamboo fiber and bamboo strips, and both achieve the effect of toughening and cracking resistance. Similarly, water treatment methods are also applicable to wood fibers. Asante et al. [109] studied the effect of hot water treatment of wood particles on the physical properties and specific compressive strength of geopolymers before and after immersion and drying. Hot water washing resulted in a reduction of 47% and 67% in the extract content of pine and eucalyptus particles, respectively, and the specific strength values of pine and eucalyptus particle geopolymers increased by 27% and 3%, respectively. Hot water pretreatment significantly increased the specific compressive strength of the pine base polymer, while the specific compressive strength of the eucalyptus base polymer did not increase. It showed that the hot water washed away the unique extracts of pine trees, so that there was better compatibility between geopolymer and wood.

## 6. Conclusions

In this paper, the larger part is the durability of CFGCs such as crack resistance and toughness, sulfate corrosion resistance, chloride ion penetration resistance, dry and wet cycle resistance, and high temperature resistance. Compared with cement-based materials, the weak alkalinity of CFGCs slows down the degradation of CFs. CFs have been widely used in CFGCs due to its excellent properties. Meanwhile, the mechanical properties of the interface between CFs and matrix should be further improved to enhance the durability of composites and lay a foundation for their engineering application. The main conclusions are as follows:All types of natural cellulose fibers can be used to reinforce geopolymers. Among the bast fibers, hemp, flax and jute, and leaf fiber sisal are the most widely used, and there is also more related research;An appropriate amount of plant fiber has a beneficial effect on the mechanical properties of the geopolymer, toughening and cracking resistance, and other types of durability. Too much mixing will have a negative effect. In CFGCs, the CF content range is mostly 0.1–10%, and the best content is usually 2–4% volume content;The alkaline degradation of CF in the geopolymer matrix has an adverse effect on the mechanical properties of the composites. Chemical modification and self-modification can be used to adjust the adhesion state of the fiber and matrix interface and optimize the properties of the interface layer between the fiber and matrix to achieve the best properties of the geopolymer;Nanomaterials can improve the microstructure of CFGCs, make the material matrix more compact, reduce the degradation rate of CF and improve the durability of CFGCs;CFGCs have good properties of resistance to sulfate and chloride ion erosion and can prevent degradation of fibers at high temperatures. However, the sugar precipitated from CFs in alkaline environment reduces the compactness of geopolymer gel and has a negative effect on its durability.

## Figures and Tables

**Figure 1 molecules-27-00796-f001:**
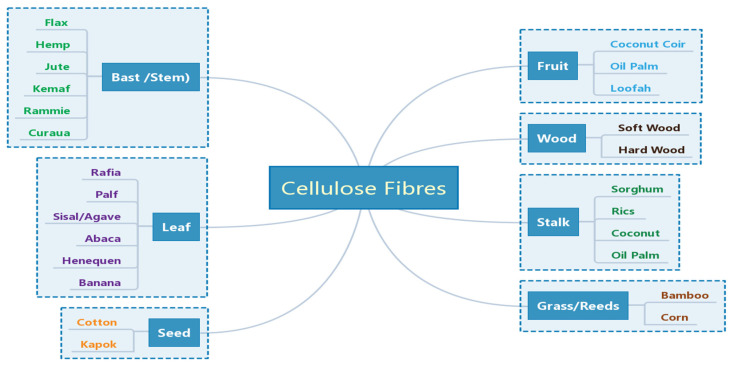
Classification of CFs used for reinforcement the geopolymers. Reprinted with permission from ref. [36]. Copyright 2020 Copyright MDPI.

**Figure 2 molecules-27-00796-f002:**
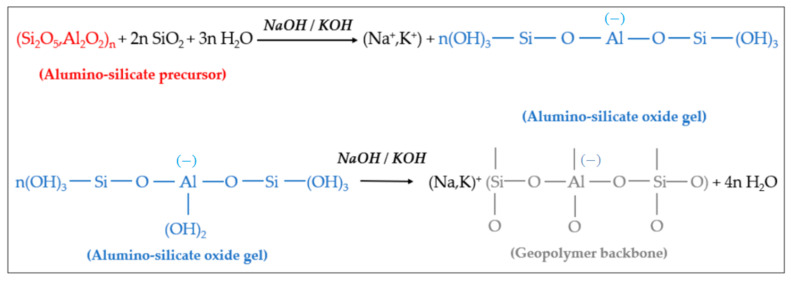
Hydrolysis and polycondensation of aluminum-silicate precursors and formation of geopolymers.

**Figure 3 molecules-27-00796-f003:**
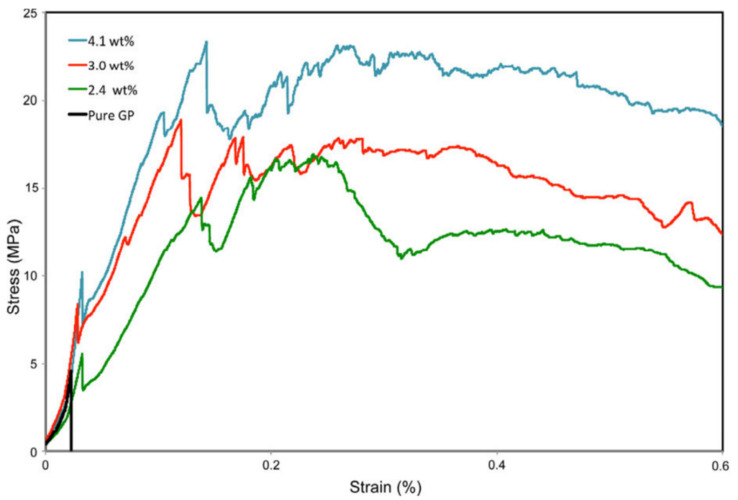
Typical stress–strain curves of pure geopolymer and composites with various fiber contents. Reprinted with permission from ref. [62]. Copyright 2015 Copyright Springer Nature.

**Figure 4 molecules-27-00796-f004:**
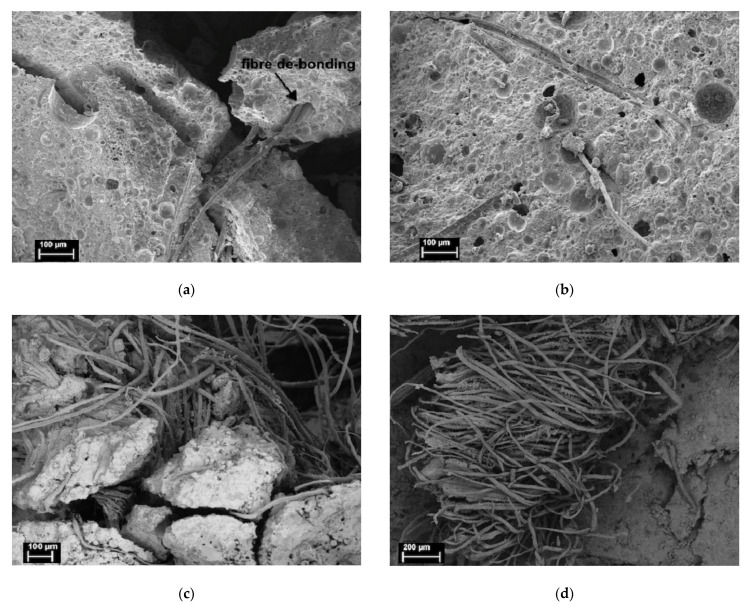
Fracture of flax-fiber-reinforced geopolymer, (**a**) fiber debonding, (**b**) fiber creasing and pulling out, (**c**) fiber bridging cracks, (**d**) fiber breaking. Reprinted with permission from ref. [62]. Copyright 2015 Copyright Springer Nature.

**Figure 5 molecules-27-00796-f005:**
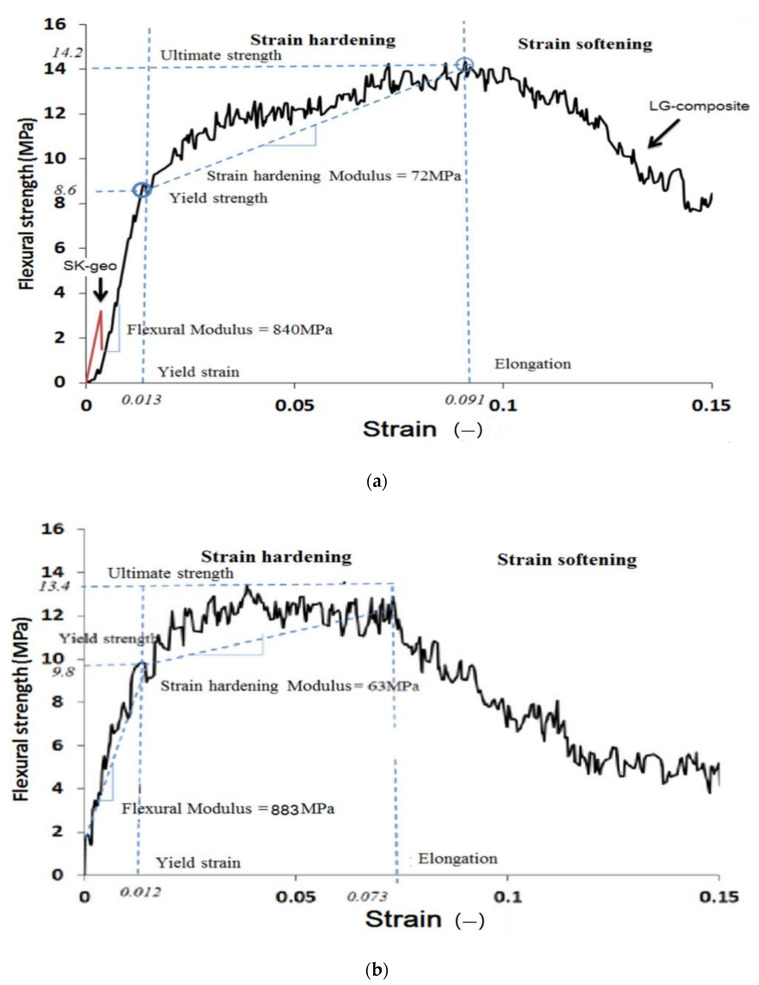
Typical stress-strain curves of geopolymer composites with the loofah fiber (10% *v*/*v*), a specimen non-aged (**a**), and a specimen aged for 20 months (**b**). Reprinted with permission from ref. [71]. Copyright 2017 Copyright Elsevier.

**Figure 6 molecules-27-00796-f006:**
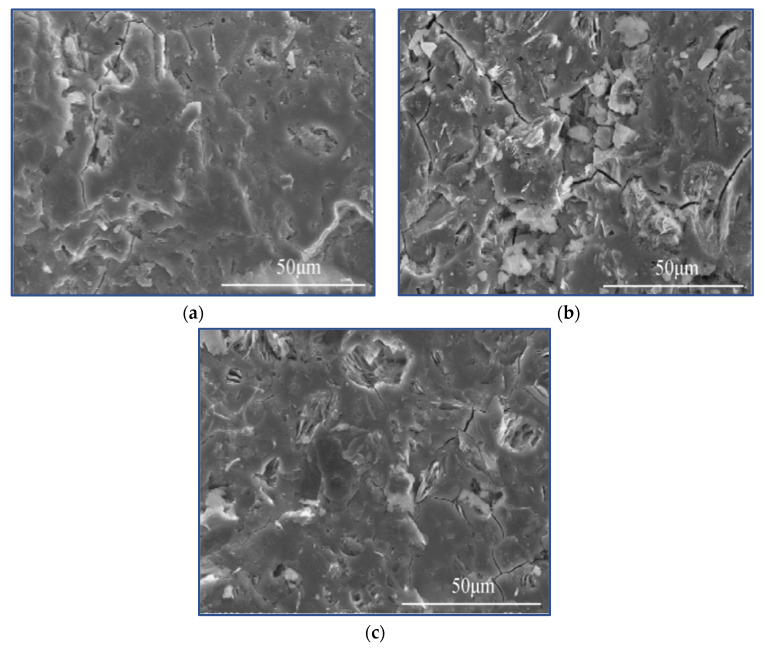
SEM images of fiber-reinforced metakaolin-based geopolymer, (**a**) before sulfate attack; (**b**) after sulfate attack at a concentration of 5 wt% and (**c**) 15 wt%. Reprinted with permission from ref. [79]. Copyright 2020 Copyright The Chinese Ceramic Society.

**Figure 7 molecules-27-00796-f007:**
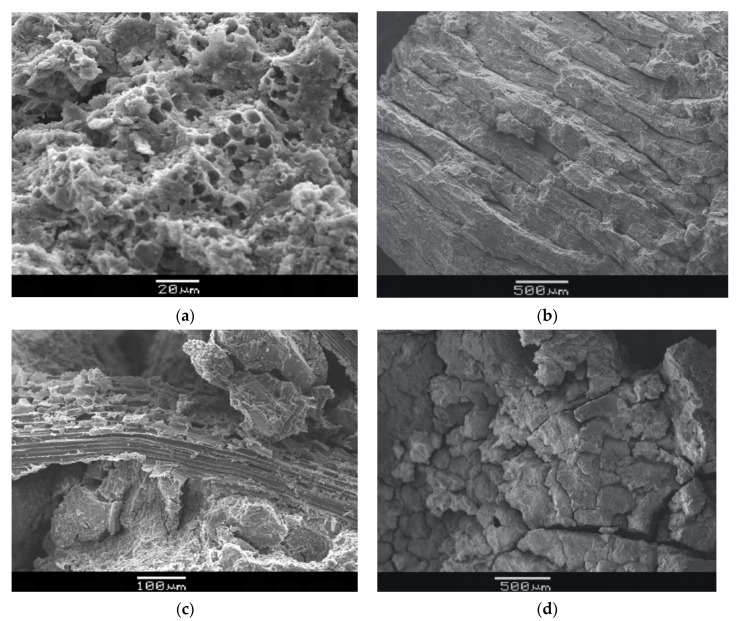
SEM micrograph of bamboo fiber-reinforced geopolymer, (**a**) acid leaching to form micro-pores; (**b**) parallel periodic micro-cracking in the geopolymer matrix; (**c**) undissolved chopped Bamboo fiber (**d**) a compound with periodic parallel microcracks. Reprinted with permission from ref. [85]. Copyright 2021 Copyright Elsevier.

**Figure 8 molecules-27-00796-f008:**
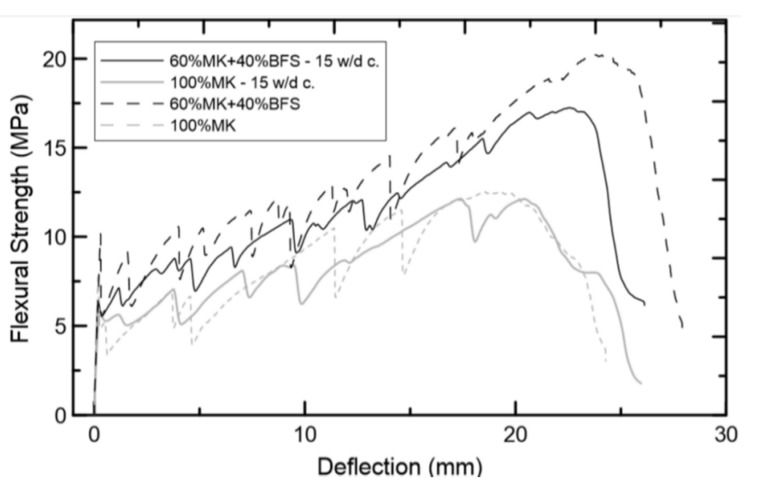
Flexural Strength–Deflection curves. Reprinted with permission from ref. [86]. Copyright 2017 Copyright The American Ceramic Society.

**Figure 9 molecules-27-00796-f009:**
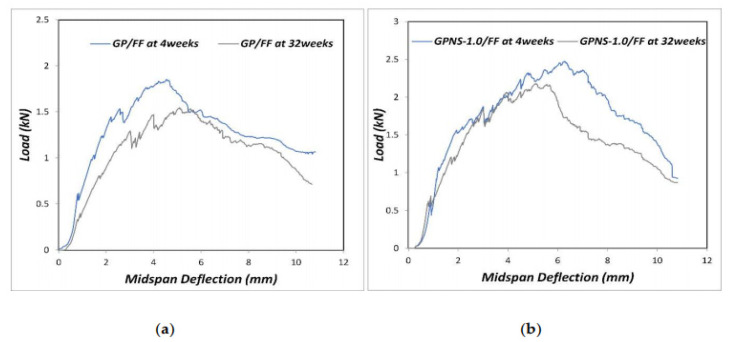
Load–deflection diagrams of (**a**) GP/FF composite and (**b**) GPNS-1/FF nanocomposite at 4 and 32 weeks. Reprinted with permission from ref. [96]. Copyright 2019 Copyright MDPI.

**Figure 10 molecules-27-00796-f010:**
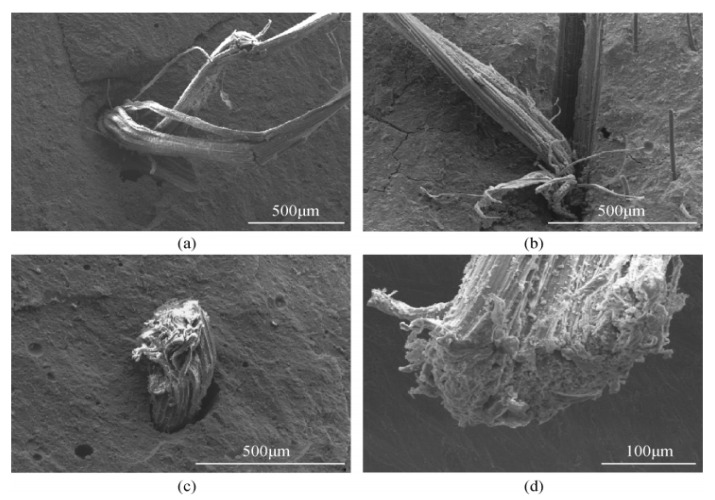
SEM images of sisal-fiber-reinforced metakaolin-based geopolymer, (**a**,**b**) the fiber is broken or torn at the section; (**c**,**d**) the damage and tear of the fiber. Reprinted with permission from ref. [77]. Copyright 2018 Copyright MDPI.

**Table 1 molecules-27-00796-t001:** Mechanical properties of typical fibers.

Fiber Type	Fiber Name	Density/ (g cm^−3^)	Tensile Strength/MPa	Specific Strength/(S ρ^−1^)	Tensile Modulus/GPa	Specific Modulus/(E ρ^−1^)	Elongation at Break/%	Ref.
Bast	Flax	1.5	800–1500	535–1000	27.6–80	18.4–53	1.2–3.2	[37]
	Hemp	1.48	550–900	372–608	70	47.3	2–4	[38]
	Jute	1.46	393–800	269–548	10–30	6.85–20.6	1.5–1.8	[39]
	Kenaf	1.45	930	641	53	36.55	1. 6	[40]
	Ramie	1.5	220–938	147–625	44–128	29.3–85	2–3.8	[41]
Leaf	Abaca	1.5	400	267	12	8	3–10	[42]
	Sisal	1.45	530–640	366–441	9.4–22	6.5–15.2	3–7	[41]
	Banana Leaf	1.35	600	444	17.85	13.2	3.36	[41]
	Coconut leaf	1.15	500	435	2. 5	2.17	20	[43]
Seed	cotton	1.6	287–597	179–373	5.5–12.6	3.44–7.9	7–8	[43]
Grass	bamboo	1.1	500	454	35.91	32.6	1.4	[43]
Fruit	Coconut shell	1.2	175	146	4–6	3.3–5	30	[41]
Wood	Soft wood	1.5	1000	667	40	26.67	4.4	[43]

**Table 2 molecules-27-00796-t002:** Strength value and water absorption test results of CFGCs.

POC/(%)	OPTF/(%)	Water Reducing Agent/(%)	Tensile/(MPa)	Shear/(MPa)	Flexural/(MPa)	Water Absorption/(%)	Cross-Section Reduction/(%)	Weight Loss/(%)
0	0	0	4.55	9.41	6.31	0.6	−1.15	−2.2
25	0	0	4.31	9.04	6.02	0.8	−1.2	−2.2
50	0	0	3.94	7.94	5.56	1.4	−1.2	−2.6
75	0	0	3.62	7.09	5.10	1.8	−1.5	−2.8
100	0	0	2.91	6.42	4.78	3.2	−2	−3.1
100	0	0.5	3.05	6.48	4.83	3	−1.9	−3
100	1	0.5	4.41	7.19	6.86	4.9	−2	−3.4
100	2	0.5	3.44	6.93	5.34	7.8	−2.6	−3.9
100	3	0.5	3.21	6.54	5.03	12.5	−3.9	−5.7

## Data Availability

Not applicable.

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
