# Peer review of "Durability of Cellulosic-Fiber-Reinforced Geopolymers: A Review"

_molecules, 2022, doi:10.3390/molecules27030796_

Round 1

Reviewer 1 Report

The presented manuscript describes a review of the durability of cellulosic fibre reinforced geopolymers. The manuscript is well written, and its aim is very significant. The used figures and tables were very helpful. Only some parts of the manuscript need improvement, i.e.
•    Figure 7 - the Fig quality is very poor and unreadable.
•    Have there been similar reviews so far? How is this manuscript different from the rest?

Author Response

Reviewer 1:

The presented manuscript describes a review of the durability of cellulosic fibre reinforced geopolymers. The manuscript is well written, and its aim is very significant. The used figures and tables were very helpful. Only some parts of the manuscript need improvement, i.e.
•    Figure 7 - the Fig quality is very poor and unreadable.
•    Have there been similar reviews so far? How is this manuscript different from the rest?

Thank you very much for reviewing our manuscript in your busy schedule. Based on your valuable suggestions, we have carefully revised our manuscript. The goal point of this manuscript is achieved successfully.
But:

  1. Figure 7 - the Fig quality is very poor and unreadable.

Thank you very much for your review. According to your suggestion, we have carefully revised the Figure 7 (Now the Figure 6). (The revised page 9 and page 10, lines 342-343)

  1. Have there been similar reviews so far? How is this manuscript different from the rest?

Thank you very much for your review. According to your suggestion, we have carefully revised the section of Introduction, and added the differences from other similar reviews. (The revised page 2, lines 72-76)

Reviewer 2 Report

The manuscript entitled "Durability of cellulosic fibre reinforced geopolymers: a review "  is a review based on 109 references. It should be emphasized that the cited works come mainly from the last few years, i.e. they cover the latest state of knowledge in the described field.

The article is written correctly and in clear language. Some minor errors do not diminish the work that in my opinion is ready for publication. 

The fragments that need improvement or correction:

line 197 
"In recent years, Many studies have been reported recently on the physical, thermal ..."

line 438

Table 2 This is a table. Tables should be placed in the main text near to the first time they are  cited

lines 541-545

"When cotton fibre was added, no cracks were found on the surface of the geopolymer at the  same temperature, indicating that the cotton fibre was very effective in preventing the  cracks caused by the high temperature formed by the small cavities formed by fibre degradation in the matrix".

The sentence is very long and its final part is confused.

Author Response

Reviewer 2:

The manuscript entitled "Durability of cellulosic fibre reinforced geopolymers: a review "  is a review based on 109 references. It should be emphasized that the cited works come mainly from the last few years, i.e. they cover the latest state of knowledge in the described field.

The article is written correctly and in clear language. Some minor errors do not diminish the work that in my opinion is ready for publication. 

The fragments that need improvement or correction:

line 197 
"In recent years, Many studies have been reported recently on the physical, thermal ..."

line 438

Table 2 This is a table. Tables should be placed in the main text near to the first time they are  cited

lines 541-545

"When cotton fibre was added, no cracks were found on the surface of the geopolymer at the  same temperature, indicating that the cotton fibre was very effective in preventing the  cracks caused by the high temperature formed by the small cavities formed by fibre degradation in the matrix".

The sentence is very long and its final part is confused.

Thank you very much for reviewing our manuscript in your busy schedule. Based on your valuable suggestions, we have carefully revised our manuscript. The goal point of this manuscript is achieved successfully. With a distinguished using of very recent references.
But:

  1. line 197 

"In recent years, Many studies have been reported recently on the physical, thermal ...".

Thank you very much for your review. According to your suggestion, we have carefully and comprehensively revised the manuscript. (The revised page 6, lines 201-202)

  1. line 438

Table 2 This is a table. Tables should be placed in the main text near to the first time they are  cited.

Thank you very much for your review. According to your suggestion, we have carefully revised the manuscript, and corrected the typos in the manuscript. (The revised page 12, lines 440)

  1. lines 541-545

"When cotton fibre was added, no cracks were found on the surface of the geopolymer at the  same temperature, indicating that the cotton fibre was very effective in preventing the  cracks caused by the high temperature formed by the small cavities formed by fibre degradation in the matrix".

The sentence is very long and its final part is confused.

Thank you very much for your review. According to your suggestion, we have carefully enriched the content. (The revised page 15, lines 548-551)

Reviewer 3 Report

Reviewer’s comments

Manuscript Number: molecules-1536540

Title: Durability of cellulosic fibre reinforced geopolymers: a review

Journal: molecules

This manuscript reviews the literature research on the durability of cellulosic fiber reinforced geopolymer composites. Actually, there are so many reviews on the same point such as; Cement and Concrete Composites 107, 2020, 103498, https://doi.org/10.1016/j.cemconcomp.2019.103498 and Composites Part B: Engineering 92, 1 2016, 94-132, https://doi.org/10.1016/j.compositesb.2016.02.002. In addition, the current version of the manuscript is close to another review (Materials 2020, 13(20), 4603; https://doi.org/10.3390/ma13204603), where Figs. 1, 3, 4 are the same in the current manuscript and the published review. Thus, the current manuscript does not carry new information.

Author Response

Reviewer 3

This manuscript reviews the literature research on the durability of cellulosic fiber reinforced geopolymer composites. Actually, there are so many reviews on the same point such as; Cement and Concrete Composites 107, 2020, 103498, https://doi.org/10.1016/j.cemconcomp.2019.103498 and Composites Part B: Engineering 92, 1 2016, 94-132, https://doi.org/10.1016/j.compositesb.2016.02.002. In addition, the current version of the manuscript is close to another review (Materials 2020, 13(20), 4603; https://doi.org/10.3390/ma13204603), where Figs. 1, 3, 4 are the same in the current manuscript and the published review. Thus, the current manuscript does not carry new information.

  1. Actually, there are so many reviews on the same point such as; Cement and Concrete Composites 107, 2020, 103498, https://doi.org/10.1016/j.cemconcomp.2019.103498

Thank you very much for reviewing our manuscript in your busy schedule. Based on your valuable suggestions, we have carefully revised our manuscript. (The revised page 2, lines 72-76). At the same time, we have carefully compared and analyzed our manuscript with this review:

(1)The title of this reviewis "Fiber-Reinforced Geopolymer Composites: A Review", which mainly focuses on the research of steel Fibre, inorganic Fibre, carbon Fibre, polymer Fibre, etc.

(2) In this review, the durability of fibre-reinforced geopolymers has not been studied in detail, and there is no research on the durability of cellulosic fibre reinforced geopolymers.

(3) The main content, research ideas, research methods and conclusions of our manuscript are completely different from this review.

  1. Composites Part B: Engineering 92, 1 2016, 94-132, https://doi.org/10.1016/j.compositesb.2016.02.002.

Thank you very much for reviewing our manuscript in your busy schedule. Based on your valuable suggestions, we have carefully revised our manuscript. (The revised page 1, lines 72-76)

At the same time, we have carefully compared and analyzed our manuscript with this review. This review has been cited in our manuscript reference [35].

(1) The title of this review is "A review of recent research on the use of cellulosic fibres, their fibre fabric reinforced cementitious, geo-polymer and polymer composites in civil engineering

(2) This review presents a summary of recent development on cellulosic fibre Fabric Reinforced Cementitious (FRC) and Fabric Reinforced Geopolymer (FRG) composites, as well as their cellulosic Fabric Reinforced Polymer (FRP) composites as reinforcements of concrete, masonry and timber structures for civil engineering applications. The mechanism of action between fabric and fibre in composite is very different.

(3) The durability of fibre-reinforced geopolymers has not been studied in detail.

  1. In addition, the current version of the manuscript is close to another review (Materials 2020, 13(20), 4603; https://doi.org/10.3390/ma13204603), where Figs. 1, 3, 4 are the same in the current manuscript and the published review. Thus, the current manuscript does not carry new information.

This review has been cited in our manuscript. This paper presents a review of the physical, chemical and biological pre-treatments that have been performed on natural fibres, their results and effects on the fibre–matrix interface of cement and geopolymer composites.

This review does not address the durability of composites.

  • Illustration of Figure 1 of our manuscript.

Figure 1 shows the conventional classification of cellulosic fibres. The authors selected some representative cellulosic fibres to complete the graph drawing by themselves.

Based on your valuable suggestions, we have carefully revised our manuscript.

(The manuscript notes: Redraw according to ref. [36] ). (The revised page 2, lines 71).

  • Figures3 and Figures 4 

Figures 3 and Figures 4 of our manuscript refer to this review [36].

Based on your valuable suggestions and the content needs of the manuscript, Figure 3 has been deleted. (The revised page 4, lines 139-141).

Figure 4 (Now the Figure 3) shows the alkaline degradation process of cellulosic fibres in cement matrix. This manuscript uses references to illustrate the alkaline degradation process of cellulosic fibres in geopolymer matrix under weak alkali conditions.(The revised page 5, lines 157-161)

Finally, thank you again for your wonderful review of our manuscript in your busy schedule.

Reviewer 4 Report

In the paper of Liu and Lv, the typical properties of cellulose fiber reinforced geopolymers are surveyed.

One of the basic problems with this ms, is that the subject is not well organized on focused.

The subject “geopolymers” should be defined explicitly earlier in the ms.

The phrase “vegetable fibre” is somewhat annoying. Could not it be “plant fibre”?

Figure 1 should be re-inserted later, after it was mentioned.

It would be desirable to show the structure of cellulose.

The readers were not spoiled with illustrations, but Fig. 3 seems to be completely trivial. Why to show the removal of lignin from among the cellulose fibers?

A part of the illustrations should be transferred into the Supplementary.

The larger part of the ms is on the properties of the different geopolymers: toughness, sulfate corrosion resistance, chloride penetration resistance, crack resistance, wet- and temperature resistance.

The other major problem is that the content (that is not on chemistry) does not match the scope of Molecules.

This referee suggests rejecting this ms in “Molecules” and reconsidering after revision in “Materials”.

Author Response

Thank you very much for reviewing our manuscript in your busy schedule. Based on your valuable suggestions, we have carefully revised our manuscript. The goal point of this manuscript is achieved successfully.

    1. One of the basic problems with this ms, is that the subject is not well organized on focused.

Thank you very much for your review. According to your suggestion, we have carefully revised our manuscript. The goal point of this manuscript is achieved successfully.

 2. The subject “geopolymers” should be defined explicitly earlier in the ms.

Thank you very much for your review. According to your suggestion, we have carefully revised the section of Introduction. (The revised page 1, lines 27-36)

  1. The phrase “vegetable fibre” is somewhat annoying. Could not it be “plant fibre”?

Thank you very much for your review. According to your suggestion, we have carefully revised the manuscript.

  1. Figure 1 should be re-inserted later, after it was mentioned.

Thank you very much for your review. According to your suggestion, we have carefully revised the manuscript. (The revised page 2-3, lines 96-99)

  1. It would be desirable to show the structure of cellulose.

Thank you very much for your review. According to your suggestion, we have carefully revised the manuscript. (The revised page 2, lines 90-95)

  1. The readers were not spoiled with illustrations, but Fig. 3 seems to be completely trivial. Why to show the removal of lignin from among the cellulose fibers?

Thank you very much for your review. Based on your valuable suggestions and the content needs of the manuscript, Figure 3 has been deleted. (The revised page 5, lines 165)

  1. A part of the illustrations should be transferred into the Supplementary.

Thank you very much for your review. According to your suggestion, we have carefully revised the manuscript. A part of the illustrations will be transferred into the Supplementary as required.

  1. The other major problem is that the content (that is not on chemistry) does not match the scope of Molecules.This referee suggests rejecting this ms in “Molecules” and reconsidering after revision in “Materials”.

Thank you very much for your review. Based on your valuable suggestions, the structure content of cellulose fiber was added.

The manuscript addresses the durability of cellulosic fibre reinforced geopolymer composites. Its durability is related to the properties of cellulose, geopolymer and interface microstructure. The manuscript content matchs to the scope of Molecular (Polymer Chemistry). 

In addition, the content has been reviewed by the editor before submission, which falls within the scope of the "Polymer Chemistry" section:-Special issue "Lignocellulosic Biomass II".

Finally, thank you again for your wonderful review of our manuscript in your busy schedule.

Round 2

Reviewer 3 Report

The author tried to address the comments on the manuscript during the revision, it is really appreciated. However, the work still does not carry new information compared with the literature. Thus, it is still not suitable for publication in molecules.

Author Response

Thank you very much for your review. Based on your valuable suggestions, we have further carefully revised the manuscript. The revised parts of the manuscript have been marked in red. Figure 3 has been deleted. (The revised page 5, lines 165)

(1)In our review manuscript, the larger part is on the durability of cellulosic fibre-reinforced geopolymers, including the degradation of fibres in geopolymer matrix, the toughness of fibre against matrix cracking, the acid resistance, and resistance to chloride ion penetration, high temperature resistance, etc.

(2)In other published reviews, there is no literature on the durability of cellulosic fibre-reinforced geopolymers. (Including toughness, sulfate corrosion resistance, chloride penetration resistance, crack resistance, wet- and temperature resistance).

(3)The main content, research ideas, research methods and conclusions of our review manuscript are completely different from other published reviews.

 The work has carried new information compared with the literature.

Finally, thank you again for your wonderful review of our manuscript in your busy schedule.